# The Effect of Brief Warming during Induction of General Anesthesia and Warmed Intravenous Fluid on Intraoperative Hypothermia in Patients Undergoing Urologic Surgery

**DOI:** 10.3390/medicina60050747

**Published:** 2024-04-30

**Authors:** Ye-Ji Oh, In-Jung Jun

**Affiliations:** Department of Anesthesiology and Pain Medicine, Sanggye Paik Hospital, Inje University College of Medicine, Seoul 01757, Republic of Korea; s4749@paik.ac.kr

**Keywords:** hypothermia, care, perioperative, body temperature regulation, urology

## Abstract

*Background and Objectives*: Transurethral urologic surgeries frequently lead to hypothermia due to bladder irrigation. Prewarming in the preoperative holding area can reduce the risk of hypothermia but disrupts surgical workflow, preventing it from being of practical use. This study explored whether early intraoperative warming during induction of anesthesia, known as peri-induction warming, using a forced-air warming device combined with warmed intravenous fluid could prevent intraoperative hypothermia. *Materials and Methods*: Fifty patients scheduled for transurethral resection of the bladder (TURB) or prostate (TURP) were enrolled and were randomly allocated to either the peri-induction warming or control group. The peri-induction warming group underwent whole-body warming during anesthesia induction using a forced-air warming device and was administered warmed intravenous fluid during surgery. In contrast, the control group was covered with a cotton blanket during anesthesia induction and received room-temperature intravenous fluid during surgery. Core temperature was measured upon entrance to the operating room (T_0_), immediately after induction of anesthesia (T_1_), and in 10 min intervals until the end of the operation (T_end_). The incidence of intraoperative hypothermia, change in core temperature (T_0_–T_end_), core temperature drop rate (T_0_–T_end_/[duration of anesthesia]), postoperative shivering, and postoperative thermal comfort were assessed. *Results*: The incidence of intraoperative hypothermia did not differ significantly between the two groups. However, the peri-induction warming group exhibited significantly less change in core temperature (0.61 ± 0.3 °C vs. 0.93 ± 0.4 °C, *p* = 0.002) and a slower core temperature drop rate (0.009 ± 0.005 °C/min vs. 0.013 ± 0.004 °C/min, *p* = 0.013) than the control group. The peri-induction warming group also reported higher thermal comfort scores (*p* = 0.041) and less need for postoperative warming (*p* = 0.034) compared to the control group. *Conclusions*: Brief peri-induction warming combined with warmed intravenous fluid was insufficient to prevent intraoperative hypothermia in patients undergoing urologic surgery. However, it improved patient thermal comfort and mitigated the absolute amount and rate of temperature drop.

## 1. Introduction

Perioperative hypothermia, defined as core temperature < 36.0 °C, can lead to a range of significant complications, such as surgical site infections, delayed wound healing, and increased myocardial oxygen demand, that raise the risk of cardiovascular complications [1]. Hypothermia commonly occurs during transurethral resection of the bladder (TURB) or prostate (TURP) due to extensive irrigation of the bladder and affects a high proportion of elderly patients [2]. These procedures are typically conducted in the lithotomy position, making it impossible to implement whole-body forced-air warming. Therefore, preventive measures are crucial to avoid hypothermia during transurethral urologic surgery.

Prewarming, which is warming the patient before anesthesia induction, with convective forced-air warming blankets prevents hypothermia by eliminating the central–peripheral temperature gradient [3,4]. Recent reports have indicated that even 10 min of prewarming can mitigate a decline in intraoperative core temperature [5,6,7]. However, existing prewarming methods occur in the preanesthesia holding area and range from as little as 10 min to as much as an hour, disrupting the operating room workflow. There is a need for an efficient method to prevent intraoperative hypothermia.

We assumed implementing early intraoperative warming during induction of anesthesia in the operating room, known as peri-induction warming, can be more time- and space-efficient than prewarming in the preanesthesia holding area. This study investigated whether brief peri-induction warming with warmed intravenous fluids can effectively prevent hypothermia.

## 2. Materials and Methods

### 2.1. Study Design and Participants

This prospective, randomized, controlled study was approved by the Institutional Review Board of Sanggye Paik Hospital (approval No. 2022-08-011-001) and was registered at clinicaltrials.gov (registration No. NCT05636189) before patient enrollment. It was conducted as a single-blinded (outcome assessor) study. Patients were enrolled from November 2022 to August 2023. Written informed consent was obtained from all patients.

Patients over the age of 20 scheduled for TURB or TURP under general anesthesia in a single university hospital were screened for eligibility. The exclusion criteria were pre-induction body temperature outside of the normal range, namely >37.5 °C or <36.0 °C; having moderate-to-severe cardiopulmonary or renal disease; having thyroid disease; suspicion or diagnosis of infection; and patient refusal. The dropout criteria were failure to follow up and operation conversion to open surgery.

Enrolled patients were randomly allocated to either the peri-induction warming or control group. A random sequence was generated using Microsoft Excel 2021, and patient group allocation information was concealed in sequentially numbered, opaque, sealed envelopes. When the patient arrived at the preanesthesia holding area, an anesthesiologist not involved in data collection opened the patient’s envelope and assigned them to the group noted inside. The outcome assessor was an anesthesia nurse not involved in anesthesia induction and did not know which group the patient was assigned to.

### 2.2. Protocol

The ambient temperature of the preanesthesia holding area, operating room, corridor, and postanesthesia care unit (PACU) was kept within 21–23 °C. When the patients arrived in the preanesthesia holding area, they were covered with a cotton blanket and then taken to the operating room. The transfer time from the preanesthesia holding area to the operating room was less than 1 min for all patients.

The patients in the peri-induction warming group were warmed with a Bairhugger model 505 forced-air warming device (Arizant Healthcare, Eden Prairie, MN, USA) set to high, which was equivalent to 43 °C, from laying on the surgical bed until anesthesia induction was completed. During peri-induction warming, the patients were covered with a WarmTouch full-body forced-air blanket (Covidien, LLC, Mansfield, MA, USA) underneath the cotton blanket, which covered them from their lower neck to their feet but did not cover the arms. The duration of peri-induction warming was not fixed for no delay in the start of operation. The patients in the peri-induction warming group were administered warmed Plasma Solution A (HK inno.N, Seoul, Republic of Korea) kept in a warming cabinet at 41 °C for more than 8 h. The patients in the control group were not warmed during anesthesia induction. They were covered with a cotton blanket that covered them from their lower neck to their feet but did not cover their arms. Plasma Solution A (HK inno.N, Seoul, Republic of Korea) kept at room temperature was administered.

All patients received 8 mL/kg of intravenous fluid during anesthesia induction followed by 2 mL/kg/hr of intravenous fluid during the operation. The rate at which fluid was administered was adjusted at the anesthesiologist’s discretion. After the patients were placed in the lithotomy position, they were covered with a COVIDIEN WarmTouch Upper Body Blanket (Covidien, LLC, Mansfield, MA, USA), and a forced-air warming device was activated and set to medium, which was equivalent to 38 °C. If the patient’s core temperature rose above 37.0 °C, the warming device was turned off, and if it fell below 35.0 °C, the device’s temperature setting was changed to high (43 °C). The patients received whole-body warming on request in the PACU. Participants with shivering in the PACU were administered meperidine at the discretion of an anesthesiologist.

All patients were monitored with standard monitoring devices, such as three-lead electrocardiography, noninvasive blood pressure devices, and pulse oximetry. Depending on the patient’s underlying medical condition, invasive blood pressure monitoring with a radial artery catheter was employed when necessary. General anesthesia was induced using balanced anesthesia. Loss of consciousness and neuromuscular block were achieved using intravenous 1–2 mg/kg of propofol and 0.6–0.8 mg/kg of rocuronium. Remifentanil was titrated using target-controlled infusion within 0.5–5 ng/mL. During surgery, anesthesia was maintained using 1.5%–2.5% of sevoflurane with the use of a bispectral index (BIS, Aspect Medical Systems Inc., Natick, MA, USA). Kinemyography (KMG, MechanoSensor™ Datex Ohmeda GE Healthcare NMT-EMG, Helsinki, Finland) was used for monitoring neuromuscular blockade. Adequate depth of anesthesia and neuromuscular blockade was maintained with a BIS of 40–60 and train-of-four (TOF) below 1 count during operation. After endotracheal intubation, anesthesia was maintained with sevoflurane and remifentanil. Mechanical ventilation was maintained with an oxygen and air mixture with a fraction of inspired oxygen of 0.5. At the end of the surgery, the patient’s neuromuscular blockade was reversed using sugammadex. The dosage of sugammadex was adjusted under the guidance of the TOF monitor. We waited until the TOF ratio recovered to 0.9 or higher for 3 consecutive measurements, the BIS reached over 90 with adequate spontaneous breathing, and the patient opened his or her eyes and nodded in response to verbal commands. The endotracheal tube was removed, and the patient was transferred to the PACU. During the transfer to the recovery room, all patients were covered with a single cotton blanket. The transfer time from the operating room to the PACU was less than 1 min for all patients.

### 2.3. Measurements

The patients’ preoperative demographics (age, sex, height, weight, body mass index, hypertension, diabetes mellitus, American Society of Anesthesiologists classification) were recorded. Perioperative characteristics (operation type: TURB or TURP, operation duration, induction duration, PACU time, crystalloid amount, bladder irrigation fluid amount, vasoconstrictor dose, estimated blood loss, ambient temperature of operating room and PACU) were recorded. The same type of thermometer was always placed in the preanesthesia holding area, operating room, corridor, and PACU, and the temperature was recorded when the patient entered each area.

Patients’ core tympanic temperatures were measured using a Thermoscan IRT tympanic thermometer (Braun, Kronberg, Germany) while the patients were awake. Three measurements were made, and the highest one was recorded. During the operation, patients’ esophageal temperature was measured using an esophageal stethoscope (Erae SI Co., Ltd., Seoul, Republic of Korea), which was inserted 28–32 cm from the upper incisors. Core temperature was measured upon entrance to the operating room (T_0_), immediately after induction of anesthesia (T_1_), in 10 min intervals until the end of the operation (T_end_), and in 10 min intervals after arriving in the PACU. The core temperatures of all patients were measured by an anesthesia nurse who did not know which group the patient was assigned to. Intraoperative hypothermia was defined based on the temperature recorded at the end of the operation. Mild hypothermia was defined as 35.0–35.9 °C, moderate hypothermia as 34.0–34.9 °C, and severe hypothermia as <34.0 °C [8].

Change in core temperature was defined as T_0_–T_end_, and the core temperature drop rate was calculated as T_0_–T_end_/[duration of anesthesia]. Incidence of shivering was assessed upon entry into the operating room, upon entry into the PACU, and at 10 min intervals thereafter. Thermal comfort was assessed upon entry into the operating room and prior to departure from the PACU. It was evaluated using a 10-point scale with 0 = extremely cold, 5 = thermally neutral and optimally comfortable, and 10 = extremely hot. Scores below 5 indicated feeling cold, and scores above 5 indicated feeling hot.

Induction duration was defined as the amount of time that elapsed from entry into the operating room to the start of upper-body warming in the lithotomy position. Peri-induction warming duration was defined as the amount of time taken for early intraoperative warming during induction of anesthesia. The unwarmed duration was defined as [induction duration]–[peri-induction warming duration].

The primary outcome was the incidence of intraoperative hypothermia. The secondary outcomes were the change in perioperative core temperature, perioperative core temperature drop rate, postoperative shivering, and postoperative thermal comfort.

### 2.4. Statistical Analysis

The sample size was calculated based on the assumption that a core temperature drop of 0.5 °C was the desired therapeutic effect. A greater drop tends to lead to hypothermia-induced complications [9]. The sample size for each group was calculated as 23 (α = 0.05, ß = 0.9) using G Power version 3.1.9.4 (Franz Faul, Universitat Kiel, Kiel, Germany). Assuming a dropout rate of 5%, 50 participants were required.

Statistical analysis was performed using SPSS version 22.0 (SPSS, Inc., Armonk, NY, USA) or SAS version 9.4 (SAS Institute, Inc., Cary, NC, USA). The normality of the distribution of the data was analyzed using Kolmogorov–Smirnov tests. The demographic and perioperative data were compared using Student’s *t*-tests for continuous variables and chi-squared tests for categorical variables. Mann–Whitney U tests or Fisher’s exact tests were used for nonparametric data analyses. The groups’ core temperature changes were compared using a linear mixed model in SAS version 9.4 (SAS Institute, Inc., Cary, NC, USA). *p*-values < 0.05 were considered statistically significant.

## 3. Results

A total of 55 patients who underwent TURB or TURP under general anesthesia were eligible for participation in this study. Five patients were excluded because they did not meet the inclusion criteria. One patient had severe aortic stenosis, and four patients had chronic kidney disease. The remaining 50 patients were randomly assigned to either the control group (*n* = 25) or the peri-induction warming group (*n* = 25). There were no patients who were lost to follow-up or discontinued in both groups. All 50 patients were analyzed (Figure 1).

The groups did not have statistically significantly different demographic (age, sex, height, weight, body mass index, hypertension, diabetes mellitus, American Society of Anesthesiologists classification) and perioperative characteristics (operation type and duration, induction duration, PACU time, crystalloid amount, bladder irrigation fluid amount, vasoconstrictor dose, estimated blood loss, ambient temperature) (Table 1). The groups also did not have statistically significantly different induction durations. The peri-induction warming duration in the peri-induction warming group was 10.5 ± 1.9 min. Unwarmed duration ([induction duration]–[peri-induction warming duration]) during the induction period was 13.0 ± 3.4 and 2.7 ± 1.3 min for the control group and peri-induction warming group, respectively. There were no differences in the amount of intravenous crystalloid and bladder irrigation fluid between the groups. There were no differences between the groups in operating room temperature (23 (22.6–23) °C vs. 23 (22–23.5) °C, *p* = 0.704 in control group and peri-induction warming group, respectively) and PACU temperature (22.7 ± 0.5 °C vs. 22.5 ± 0.6 °C, *p* = 0.190 in control group and peri-induction warming group, respectively).

The groups did not have significantly different incidences of hypothermia (12% vs. 6%, *p* = 0.077) (Table 2). The severity of hypothermia was not significantly different between the groups (*p* =0.185). However, there was a significant difference in changes in core body temperature and core temperature drop rate. The peri-induction warming group’s core temperature dropped less than that of the control group (0.61 ± 0.3 °C vs. 0.93 ± 0.4 °C, *p* = 0.002). The peri-induction warming group also had a slower core temperature drop rate than the control group (0.009 ± 0.005 °C/min vs. 0.013 ± 0.004 °C/min, *p* = 0.013).

The groups’ core temperatures were not statistically significantly different between the groups after anesthesia induction and throughout the PACU stay (Figure 2). There was no difference between the groups’ core temperatures on arrival in the operating room (T_0_) (36.96 ± 0.2 °C vs. 36.78 ± 0.3 °C, *p* = 0.051). There was no difference between groups’ core temperatures 60 min after induction of anesthesia (35.94 ± 0.4 °C vs. 36.1 ± 0.4 °C, *p* = 0.137). Both groups’ core temperatures were significantly lower after anesthesia was induced than before it was induced (T_0_) with *p*-value <0.001.

One patient from the control group experienced shivering in the PACU. The control group had lower thermal comfort scores (two felt severely cold, four felt moderately cold) than the peri-induction warming group (*p* = 0.041), causing them to need more warming than the control group (8% vs. 2%, *p* = 0.034) (Table 3).

## 4. Discussion

Peri-induction warming with warmed intravenous fluid administration did not reduce the incidence of intraoperative hypothermia. However, it improved patient thermal comfort and reduced the absolute amount and rate of core temperature decline. This study was significant as it first evaluated the effect of brief peri-induction warming in patients undergoing short urologic operations.

Inconsistent with our results, Cho et al. (2016) found that peri-induction warming decreased intraoperative hypothermia in patients undergoing off-pump coronary artery bypass surgery [10]. Another study on off-pump coronary artery bypass surgery also showed that warming during anesthesia induction decreased the incidence of intraoperative hypothermia at a reasonable cost without delaying the operation schedule [11]. Yoo et al. (2021) found that peri-induction warming reduced the incidence and severity of hypothermia during major operations [12]. Given that these studies had longer anesthesia induction periods than our study, their peri-induction warming durations (35 ± 6, 49.7 ± 9.9, and 20.2 ± 8.7 min, respectively) were longer than ours (10.5 ± 1.9 min), which may have contributed to this difference in results.

In our recent study, prewarming for 10 min in the preoperative holding area maintained the core temperature [7]. Thus, we hypothesized that peri-induction warming would also reduce the incidence of intraoperative hypothermia by eliminating unwarmed duration, but it did not. Three points may have contributed to differences in their outcomes.

First, the peripheral compartment was not adequately warmed in the peri-induction group. Patients’ arms were spread out on the armboards of the operating bed and were not warmed during the induction period. Peripheral compartments, including legs and arms, account for 48% of total body mass, so maintaining adequate peripheral component temperature is important for preventing redistribution hypothermia [13,14]. Warming the periphery rather than the core was reported to be an effective method for maintaining core temperature [15].

Second, the duration of peri-induction warming may be insufficient. The actual duration of peri-induction warming was not standardized, so eight patients received peri-induction warming for less than 10 min, seven of whom fell into hypothermia at the end of the operation.

Third, propofol-induced vasodilation during peri-induction warming may have hindered the preservation of peripheral heat content. Propofol was administered midway through peri-induction warming. Propofol inhibits tonic vasoconstriction and exacerbates the redistribution of body heat from the core to the periphery [16]. Ikeda et al. (1999) have demonstrated that a brief period of propofol-induced vasodilation during anesthesia induction causes a substantial redistribution hypothermia that persists throughout surgery [16]. The anesthesia induction period, during which propofol is commonly utilized, marks a critical phase characterized by rapid core-to-periphery redistribution. At this stage, any heat transfer that has already occurred to peripheral tissues cannot be restored. Therefore, to maintain core temperature, it is advisable to sufficiently elevate peripheral heat content before vasodilation occurs due to propofol. In the present study, it was determined that propofol-induced vasodilation occurred without establishing sufficient peripheral heat reserve; therefore, brief peri-induction warming could not prevent heat redistribution. We recommend peri-induction warming of at least 10 min, along with additional measures to achieve similar effects to those of 10 min prewarming. Further research on the timing of early peri-induction warming initiation is warranted.

Peri-induction warming combined with warmed intravenous fluid administration was effective at increasing postoperative thermal comfort and reducing the core temperature drop rate. The National Institute for Health and Care Excellence provides warming guidelines for patients undergoing major or intermediate surgery, but it offers no definite warming guidelines for short operations of less than 30 min [17]. The administration of warmed intravenous fluid acts as an active warming method distinct from convective warming and efficiently helps to maintain the body temperature without additional equipment or significant costs. Campbell et al. (2015) have reported that patients receiving warmed intravenous fluid showed 0.5 °C higher core temperature compared to patients receiving room-temperature intravenous fluid [18]. Sari et al. (2021) found that the infusion of unwarmed fluid greater than 1 L significantly increased the incidence of hypothermia [19]. Administering 1 L of room-temperature intravenous fluid reduced the body temperature by 0.25 °C [20]. We suggest that adding warmed intravenous fluids would be a simple and cost-effective adjunctive warming method to maximize the effect of brief peri-induction warming.

Peri-induction warming is more time- and space-efficient compared to prewarming. Prewarming entails patients spending more time in the preanesthesia holding area than usual, often with additional staff assistance for warming procedures. In contrast, peri-induction warming occurs as part of the routine induction process. Peri-induction warming does not consume additional time or space, and it does not delay the induction duration, as shown in our result (control group: 13.0 ± 3.4., peri-induction warming group: 13.2 ± 2.0, *p*-value = 0.853). As institutions would not desire surgical delays for prewarming purposes, there is a need to advance towards effective peri-induction warming methods. We believe our study holds significant value in providing a foundation for simple and effective warming methods with minimal effort for short surgeries.

### Limitations

The present study had three limitations. The first limitation was the use of a tympanic thermometer to measure the core temperature when patients were awake. The gold standard for measuring core temperature involves using a pulmonary artery catheter, but this method is impractical and highly invasive [21,22]. In the present study, to minimize errors, the tympanic temperature was measured three times, and the highest recorded temperature was utilized. The second limitation was that each patient underwent peri-induction warming for a different duration. This method was intentionally used to avoid interrupting the operating room workflow. The third limitation was the low statistical power due to the small sample size. Low power increases the risk of type 2 errors and type 1 errors with an exaggerated effect. We acknowledge the importance of statistical power in ensuring the reliability of research findings, and we are planning future studies with larger sample sizes to validate our findings and enhance the robustness of the conclusions.

## 5. Conclusions

Compared to previous studies on prewarming or prolonged peri-induction warming, this study first evaluated the effect of brief peri-induction warming combined with intravenous warmed fluid administration on intraoperative hypothermia during TURB and TURP surgeries. Brief peri-induction warming with warmed intravenous fluid is a time- and space-efficient warming method that could be effective for all surgeries, including short urologic procedures. While the results did not show a significant effect in preventing intraoperative hypothermia, they showed improved patient thermal comfort and the mitigation of the absolute amount and rate of temperature drop. We expect positive results with a more sufficient peri-induction warming period, which should be verified in further research.

## Figures and Tables

**Figure 1 medicina-60-00747-f001:**
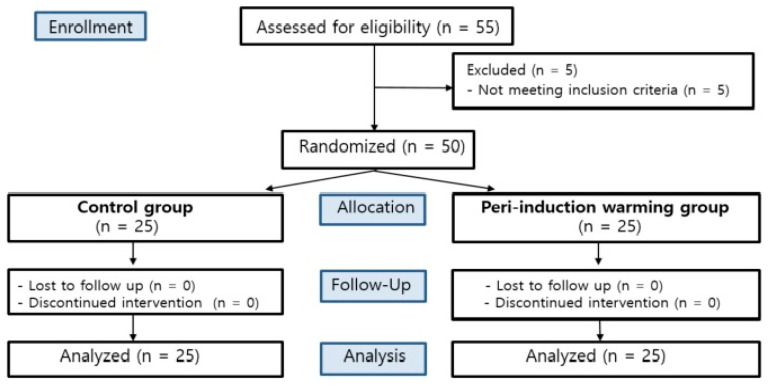
CONSORT flow gram.

**Figure 2 medicina-60-00747-f002:**
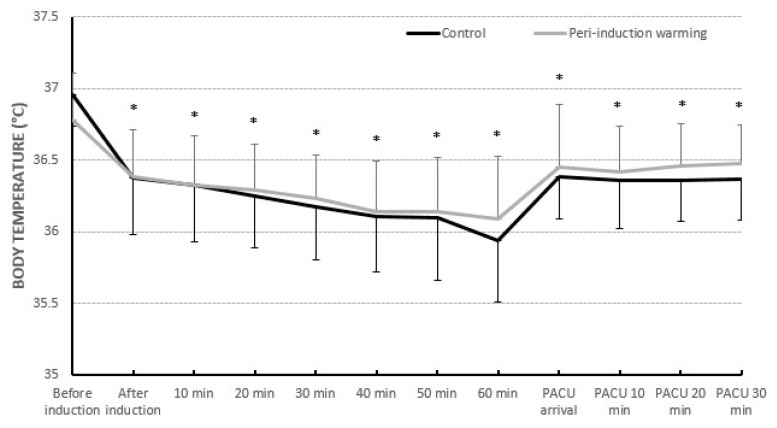
Comparison of change in core temperature. The bars are in mean ± standard deviation, * difference between times compared to baseline (T_0_: before anesthesia induction) for each group with *p* < 0.05.

**Table 1 medicina-60-00747-t001:** Demographic and perioperative data.

	Control Group (*n* = 25)	Peri-Induction Warming Group (*n* = 25)	*p*
Age (years)	67.7 ± 7.9	68.7 ± 11.4	0.710
Sex (male/female)	21/4	21/4	1.000
Height (cm)	165.2 ± 6.8	165.2 ± 8.7	0.991
Weight (kg)	67.2 ± 11.5	67.0 ± 13.2	0.974
Body mass index (kg/m^2^)	25.0 ± 3.5	24.3 ± 3.0	0.416
Hypertension	10 (40)	15 (60)	0.157
Diabetes mellitus	10 (40)	7 (28)	0.370
ASA classification			0.157
II	18 (72)	22 (88)	
III	7 (28)	3 (12)	
Operation type			0.684
TURB/TURP	21/4	22/3	
Operation duration (min)	40.9 ± 21.9	35.6 ± 29.8	0.472
Anesthesia duration (min)	76.3 ± 23.0	73.5 ± 30.4	0.711
Induction duration (min)	13.0 ± 3.4	13.2 ± 2.0	0.853
Peri-induction warming duration (min)	0	10.5 ± 1.9	0.0001
Unwarmed duration (min)	13.0 ± 3.4	2.7 ± 1.3	0.0001
PACU time (min)	45.0 ± 12.1	40.1 ± 10.0	0.127
Crystalloid amount (mL)	612.0 ± 116.0	616.8 ± 143.3	0.897
Phenylephrine dose (μg)	286 ± 389	172 ± 255	0.226
Ephedrine dose (mg)	1.8 ± 3.5	3.2 ± 4.1	0.197
Irrigation fluid amount (mL)	11024 ± 10905	11596 ± 13924	0.872
Estimated blood loss (mL)	10 (5–30)	10 (5–10)	0.351
OR temperature (°C)	23 (22.6–23)	23 (22–23.5)	0.704
PACU temperature (°C)	22.7 ± 0.5	22.5 ± 0.6	0.190

Data are shown as mean ± standard deviation or median (interquartile range) or number (%). ASA: American Society of Anesthesiologists, TURB/TURP: transurethral resection of bladder/prostate, OR: operating room, PACU: postanesthesia care unit.

**Table 2 medicina-60-00747-t002:** Comparison of core temperature between two groups.

	Control Group (*n* = 25)	Peri-Induction Warming Group (*n* = 25)	95% CI	*p*
Hypothermia	12 (48)	6 (24)		0.077
Hypothermia severity (mild/moderate/severe)	11/1/0	5/1/0		0.185
Change in core temperature (°C)	0.93 ± 0.4	0.61 ± 0.3	0.12 to 0.52	0.002
Core temperature drop rate (°C/min)	0.013 ± 0.004	0.009 ± 0.005	0.002 to 0.007	0.013

Data are shown as mean ± standard deviation, or number (%). Change in core temperature = T_0_-Tend, T_0_: tympanic temperature measured in the preoperative holding area, Tend: core temperature measured at the end of the operation, Core temperature drop rate = mean core temperature drop/anesthesia duration.

**Table 3 medicina-60-00747-t003:** Postoperative outcomes.

	Control Group (*n* = 25)	Peri-Induction Warming Group (*n* = 25)	*p*
Shivering	1 (4)	0 (0)	0.312
Warming in PACU	8 (32)	2 (8)	0.034
Thermal comfort			0.041
2 (severely cold)	2	0	
3 (moderately cold)	4	0	
4 (mildly cold)	6	4	
5 (neutral)	13	21	

Data are shown as number (%).

## Data Availability

The data are presented within the article. Additional data are available on request from the corresponding author.

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
