# Peer review of "The Effect of Brief Warming during Induction of General Anesthesia and Warmed Intravenous Fluid on Intraoperative Hypothermia in Patients Undergoing Urologic Surgery"

_medicina, 2024, doi:10.3390/medicina60050747_

Round 1

Reviewer 1 Report

Comments and Suggestions for Authors

This is a well-written and referenced small clinical research study. However, power is low (Low power = unreliable studyAn underpowered study increases the risk of type 2 errors (false negatives), but it also may increase the risk of type 1 errors as well (false positives), with an exaggerated effect. I recommend that the authors include a separate heading, "Limitations" section, and include their observations, in this regard, in that section.

Author Response

" For your convenience, I have also uploaded a PDF file containing the same content. Please refer to it as needed."

Thank you very much for the thoughtful and thorough review. We hope our responses sufficiently answered your concerns. For any points insufficient, please let me know at any time. 

Point 1. This is a well-written and referenced small clinical research study. However, power is low (Low power = unreliable study. An underpowered study increases the risk of type 2 errors (false negatives), but it also may increase the risk of type 1 errors as well (false positives), with an exaggerated effect. I recommend that the authors include a separate heading, "Limitations" section, and include their observations, in this regard, in that section.

Response 1: Thank you for your feedback. We also agree with the importance of statistical power in ensuring the reliability of the result. For this concern, we are planning to address the issue of low power in next study by increasing the sample size or making adjustments to our study design to enhance statistical power. Additionally, we will carefully consider the potential implications of both type 1 and type 2 errors in our data interpretation. Thank you for bringing this to our attention, and we added this as third limitation in a separated limitation section as follows.

# page8

4.1. Limitations

The present study had three limitations. The first limitation was use of a tympanic thermometer to measure the core temperature when patients were awake. The gold standard for measuring core temperature involves using a pulmonary artery catheter, but this method is impractical and highly invasive [21,22]. In the present study, to minimize errors, the tympanic temperature was measured three times and the highest recorded temperature was utilized. The second limitation was that each patient underwent peri-induction warming for a different duration. This method was intentionally used not to interrupt the operating room workflow. The third limitation was the low statistical power due to the small sample size. Low power increases the risk of type 2 errors and type 1 errors with an exaggerated effect. We acknowledge the importance of statistical power in ensuring the reliability of research findings that we are planning future studies with larger sample sizes to validate our findings and enhance the robustness of the conclusions.

Reviewer 2 Report

Comments and Suggestions for Authors

The manuscript is adequately presented, congratulations to the authors, but it does not represent any novelty. Also, the conclusion of the manuscript does not have a specific weight based on which it would differ from similar manuscripts on this topic. 

A special complaint is in the literature review. It is unacceptable that out of the total number of 45 references, only two are from the last 5 years. This does not represent an adequate literature review, nor can any up-to-date message, conclusion, and recommendation for readers and other authors be made.

Author Response

 "For your convenience, I have also uploaded a PDF file containing the same content. Please refer to it as needed."

Thank you very much for the thoughtful and thorough review. We hope our responses sufficiently answered your concerns. For any points insufficient, please let me know at any time. 

Point 1. The manuscript is adequately presented, congratulations to the authors, but it does not represent any novelty. Also, the conclusion of the manuscript does not have a specific weight based on which it would differ from similar manuscripts on this topic.

Response 1: Thank you for your comment. Our research began with a consideration of practical warming methods applicable in everyday clinical practice. In contrast to the extensively studied prewarming methods, peri-induction warming remains relatively unexplored. We introduced peri-induction warming during anesthesia induction, a method to be both time and space efficient compared to prewarming in the preanesthesia holding area. This study marks the first investigation using peri-induction warming with warmed fluid targeting short urologic surgeries, which are brief in duration but can precipitate significant drops in body temperature. While our findings showed limited efficacy in reducing hypothermia incidence within our study cohort, this was effective in other secondary outcome measures. We believe our research offers significant value by establishing a foundation for determining the optimal timing and duration of peri-induction warming for effective implementation in routine clinical practice.

 As your recommendation, we revised the conclusion to emphasize the differences from previous research. Also, we have added a paragraph at the end of the discussion section to highlight our novelty and its potential implications in clinical practice. They are as follows.

# page 8

  1. Conclusions

Compared to previous studies on prewarming or prolonged peri-induction warming, this study first evaluated the effect of brief peri-induction warming combined with intravenous warmed fluid administration on intraoperative hypothermia during TURB and TURP surgeries. Brief peri-induction warming with warmed intravenous fluid is time and space efficient warming method which could be effective for all surgeries, including short urologic procedures. While the results did not show a significant effect on preventing intraoperative hypothermia, they improved patient thermal comfort and mitigated the absolute amount and rate of temperature drop. We expect positive results with more sufficient peri-induction warming period, which should be verified on further research.

# page 8

Peri-induction warming is more time and space efficient compared to prewarming. Prewarming entails patients to spend more time in the preanesthesia holding area than usual, often with additional staff assistance for warming procedures. In contrast, peri-induction warming occurs as part of the routine induction process. Peri-induction warming does not consume additional time or space, and it does not delay the induction duration as shown in our result (control group: 13.0 ± 3.4., peri-induction warming group: 13.2 ± 2.0, p-value = 0.853). Any instituitions would not desire surgical delays for prewarming purposes, there is a need to advance towards effective peri-induction warming methods. We believe our study holds significant value in providing a foundation for simple and effective warming methods with minimal effort for short surgeries.

Point 2. A special complaint is in the literature review. It is unacceptable that out of the total number of 21 references, only two are from the last 5 years. This does not represent an adequate literature review, nor can any up-to-date message, conclusion, and recommendation for readers and other authors be made.

Response 2: We agree with your opinion. We have updated all the references that can be changed to the latest one as follows. I updated the references in the manuscript to ensure accuracy, and the revised references are as follows. By doing so, 12 out of the total 22 references are from papers published within the last 5 years. Reference [16] (Ikeda, 1999) remains unchanged due to its extensive citation in recent literature and a recent replacement could not be found.

[1] DiLorenzo, A.N.; Schell, R.M. Morgan & Mikhail’s clinical anesthesiology. Anesthesia & Analgesia 2014, 119, 495-496.

=> [1] Simegn, G.D.; Bayable, S.D.; Fetene, M.B. Prevention and management of perioperative hypothermia in adult elective surgical patients: A systematic review. Ann Med Surg (Lond) 2021, 72, 103059, doi:10.1016/j.amsu.2021.103059.

[2] Rabke, H.B.; Jenicek, J.A.; Khouri, E. Hypothermia associated with transurethral resection of the prostate. The Journal of Urology 1962, 87, 447-449.

=> [2] Akelma, F.K.; Ergil, J.; Özkan, D.; Arık, E.; Akkuş, İ.B.; Aydın, G.B. The effect of preoperative warming on perioperative hypothermia in transurethral prostatectomies. GUlhane Tip Dergisi 2020, 62, 114.

[3] Sessler, D.I.; Schroeder, M.; Merrifield, B.; Matsukawa, T.; Cheng, C. Optimal duration and temperature of prewarming. The Journal of the American Society of Anesthesiologists 1995, 82, 674-681.

=> [3] Ucak, A.; Tat Catal, A.; Karadag, E.; Cebeci, F. The Effect of Prewarming on Perioperative Hypothermia: A Systematic Review and Meta-analysis of Randomized Controlled Studies. J Perianesth Nurs 2024, doi:10.1016/j.jopan.2023.11.003.

[4]          Torossian, A. Thermal management during anaesthesia and thermoregulation standards for the prevention of inadvertent perioperative hypothermia. Best practice & research Clinical anaesthesiology 2008, 22, 659-668.

=> [4] Oh, E.J.; Han, S.; Lee, S.; Choi, E.A.; Ko, J.S.; Gwak, M.S.; Kim, G.S. Forced-air prewarming prevents hypothermia during living donor liver transplantation: a randomized controlled trial. Sci Rep 2023, 13, 3713, doi:10.1038/s41598-022-23930-2.

[5] Kaufner, L.; Niggemann, P.; Baum, T.; Casu, S.; Sehouli, J.; Bietenbeck, A.; Boschmann, M.; Spies, C.; Henkelmann, A.; von Heymann, C. Impact of brief prewarming on anesthesia-related core-temperature drop, hemodynamics, microperfusion and postoperative ventilation in cytoreductive surgery of ovarian cancer: a randomized trial. BMC anesthesiology 2019, 19, 1-10.

=> [5] Kawanishi, R.; Honda, Y.; Bando, Y.; Kakuta, N.; Tanaka, K. Effect of 10-minute prewarming plus intraoperative co-warming on core temperature maintenance during breast surgery compared to intraoperative co-warming alone: a randomized controlled trial. The Journal of Medical Investigation 2023, 70, 74-79.

[9]     Winkler, M.; Akca, O.; Birkenberg, B.; Hetz, H.; Scheck, T.; Arkilic, C.F.; Kabon, B.; Marker, E.; Grubl, A.; Czepan, R.; et al. Aggressive warming reduces blood loss during hip arthroplasty. Anesth Analg 2000, 91, 978-984, doi:10.1097/00000539-200010000-00039.

=> [9] Sessler, D.I.; Nault, R.J. Perioperative Temperature Monitoring: Reply. Anesthesiology 2021, 135, 190, doi:10.1097/ALN.0000000000003793.

[13] Yamakage, M.; Kamada, Y.; Honma, Y.; Tsujiguchi, N.; Namiki, A. Predictive variables of hypothermia in the early phase of general anesthesia. Anesth Analg 2000, 90, 456-459, doi:10.1097/00000539-200002000-00040.

=> [13] Roth, J.V. Techniques to Reduce the Magnitude and Duration of Redistribution Hypothermia in Adults. In Autonomic Nervous System Monitoring-Heart Rate Variability; IntechOpen: 2020.

 [21]  Stavem, K.; Saxholm, H.; Smith-Erichsen, N. Accuracy of infrared ear thermometry in adult patients. Intensive care medicine 1997, 23, 100-105.

=> [22] Hymczak, H.; Golab, A.; Mendrala, K.; Plicner, D.; Darocha, T.; Podsiadlo, P.; Hudziak, D.; Gocol, R.; Kosinski, S. Core Temperature Measurement-Principles of Correct Measurement, Problems, and Complications. Int J Environ Res Public Health 2021, 18, doi:10.3390/ijerph182010606.

Round 2

Reviewer 2 Report

Comments and Suggestions for Authors

The authors corrected their manuscript in accordance with the recommendations